# SCALABLE UNBALANCED OPTIMAL TRANSPORT USING GENERATIVE ADVERSARIAL NETWORKS

**Karren D. Yang & Caroline Uhler**
Laboratory for Information & Decision Systems
Institute for Data, Systems and Society
Massachusetts Institute of Technology
Cambridge, MA, USA
`{karren, cuhler}@mit.edu`

## ABSTRACT

Generative adversarial networks (GANs) are an expressive class of neural generative models with tremendous success in modeling high-dimensional continuous measures. In this paper, we present a scalable method for unbalanced optimal transport (OT) based on the generative-adversarial framework. We formulate unbalanced OT as a problem of simultaneously learning a transport map and a scaling factor that push a source measure to a target measure in a cost-optimal manner. We provide theoretical justification for this formulation, showing that it is closely related to an existing static formulation by Liero et al. (2018). We then propose an algorithm for solving this problem based on stochastic alternating gradient updates, similar in practice to GANs, and perform numerical experiments demonstrating how this methodology can be applied to population modeling.

## 1 INTRODUCTION

We consider the problem of unbalanced optimal transport: given two measures, find a cost-optimal way to transform one measure to the other using a combination of mass variation and transport. Such problems arise, for example, when modeling the transformation of a source population into a target population (Figure 1a). In this setting, one needs to model mass transport to account for the features that are evolving, as well as local mass variations to allow sub-populations to become more or less prominent in the target population (Schiebinger et al., 2017).

Classical optimal transport (OT) considers the problem of pushing a source to a target distribution in a way that is optimal with respect to some transport cost without allowing for mass variations. Modern approaches are based on the Kantorovich formulation (Kantorovich, 1942), which seeks the optimal probabilistic coupling between measures and can be solved using linear programming methods for discrete measures. Recently, Cuturi (2013) showed that regularizing the objective using an entropy term allows the dual problem to be solved more efficiently using the Sinkhorn algorithm. Stochastic methods based on the dual objective have been proposed for the continuous setting (Genevay et al., 2016; Seguy et al., 2017; Arjovsky et al., 2017). Optimal transport has been applied to many areas, such as computer graphics (Ferradans et al., 2014; Solomon et al., 2015) and domain adaptation (Courty et al., 2014; 2017).

In many applications where a transport cost is not available, transport maps can also be learned using generative models such as generative adversarial networks (GANs) (Goodfellow et al., 2014), which push a source distribution to a target distribution by training against an adversary. Numerous transport problems in image translation (Mirza & Osindero, 2014; Zhu et al., 2017; Yi et al., 2017), natural language translation (He et al., 2016), domain adaptation (Bousmalis et al., 2017) and biological data integration (Amodio &

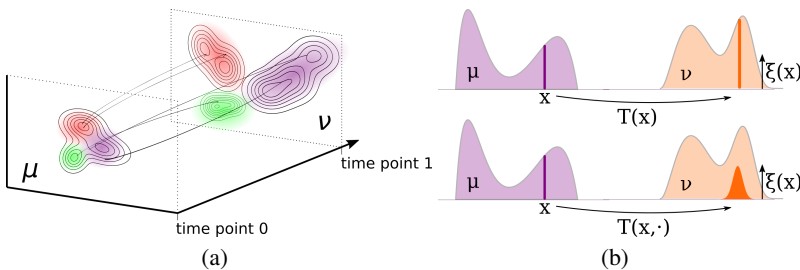

Figure 1: (a) Illustration of the problem of modeling the transformation of a source population $\mu$ to a target population $\nu$. In this example, one sub-population is growing more rapidly than the others. (b) Schematic of Monge-like formulations of unbalanced optimal transport. The objective is to learn a transport map $T$ (for transporting mass) and scaling factor $\xi$ (for mass variation) to push the source $\mu$ to the target $\nu$, using a deterministic transport map (top) (Chizat et al., 2015) or a stochastic transport map (bottom).

Krishnaswamy, 2018) have been tackled using variants of GANs, with strategies such as conditioning or cycle-consistency employed to enforce correspondence between original and transported samples. However, all these methods conserve mass between the source and target and therefore cannot handle mass variation.

Several formulations have been proposed for extending the theory of OT to the setting where the measures can have unbalanced masses (Chizat et al., 2015; 2018; Kondratyev et al., 2016; Liero et al., 2018; Frogner et al., 2015). In terms of numerical methods, a class of scaling algorithms (Chizat et al., 2016) that generalize the Sinkhorn algorithm for balanced OT have been developed for approximating the solution to *optimal entropy-transport* problems; this formulation of unbalanced OT by Liero et al. (2018) corresponds to the Kantorovich OT problem in which the hard marginal constraints are relaxed using divergences to allow for mass variation. In practice, these algorithms have been used to approximate unbalanced transport plans between discrete measures for applications such as computer graphics (Chizat et al., 2016), tumor growth modeling (Chizat & Di Marino, 2017) and computational biology (Schiebinger et al., 2017). However, while optimal entropy-transport allows mass variation, it cannot explicitly model it, and there are currently no methods that can perform unbalanced OT between *continuous* measures.

**Contributions.** Inspired by the recent successes of GANs for high-dimensional transport problems, we present a novel framework for unbalanced optimal transport that directly models mass variation in addition to transport. Concretely, our contributions are the following:

- We propose to solve a Monge-like formulation of unbalanced OT, in which the goal is to learn a stochastic transport map and scaling factor to push a source to a target measure in a cost-optimal manner. This generalizes the unbalanced Monge OT problem by Chizat et al. (2015).

- By relaxing this problem, we obtain an alternative form of the optimal entropy-transport problem by Liero et al. (2018), which confers desirable theoretical properties.

- We develop scalable methodology for solving the relaxed problem. Our derivation uses a convex conjugate representation of divergences, resulting in an alternating gradient descent method similar to GANs (Goodfellow et al., 2014).

- We demonstrate in practice how our methodology can be applied towards population modeling using the MNIST and USPS handwritten digits datasets, the CelebA dataset, and a recent single-cell RNA-seq dataset from zebrafish embrogenesis.

In addition to these main contributions, for completeness we also propose a new scalable method (Algorithm 2) in the Appendix for solving the optimal-entropy transport problem by Liero et al. (2018) in the continuous

setting. The algorithm extends the work of Seguy et al. (2017) to unbalanced OT and is a scalable alternative to the algorithm of Chizat et al. (2016) for very large or continuous datasets.

## 2 PRELIMINARIES

**Notation.** Let $\mathcal{X}, \mathcal{Y} \subseteq \mathbb{R}^n$ be topological spaces and let $\mathcal{B}$ denote the Borel $\sigma$-algebra. Let $\mathscr{M}_+^1(\mathcal{X}), \mathscr{M}_+(\mathcal{X})$ denote respectively the space of probability measures and finite non-negative measures over $\mathcal{X}$. For a measurable function $T$, let $T_\#$ denote its pushforward operator: if $\mu$ is a measure, then $T_\#\mu$ is the pushforward measure of $\mu$ under $T$. Finally, let $\pi^{\mathcal{X}}$, $\pi^{\mathcal{Y}}$ be functions that project onto $\mathcal{X}$ and $\mathcal{Y}$; for a joint measure $\gamma \in \mathscr{M}_+(\mathcal{X} \times \mathcal{Y})$, $\pi_\#^{\mathcal{X}}\gamma$ and $\pi_\#^{\mathcal{Y}}\gamma$ are its marginals with respect to $\mathcal{X}$ and $\mathcal{Y}$ respectively.

**Optimal transport (OT)** addresses the problem of transporting between measures in a cost-optimal manner. Monge (1781) formulated this problem as a search over deterministic transport maps. Specifically, given $\mu \in \mathscr{M}_+^1(\mathcal{X}), \nu \in \mathscr{M}_+^1(\mathcal{Y})$ and a cost function $c : \mathcal{X} \times \mathcal{Y} \to \mathbb{R}^+$, Monge OT seeks a measurable function $T : \mathcal{X} \to \mathcal{Y}$ minimizing

$$\inf_T \int_{\mathcal{X}} c(x, T(x)) \, d\mu(x) \tag{1}$$

subject to the constraint $T_\#\mu = \nu$. While the optimal $T$ has an intuitive interpretation as an optimal transport map, the Monge problem is non-convex and not always feasible depending on the choices of $\mu$ and $\nu$. The *Kantorovich OT* problem is a convex relaxation of the Monge problem that formulates OT as a search over probabilistic transport plans. Given $\mu \in \mathscr{M}_+^1(\mathcal{X}), \nu \in \mathscr{M}_+^1(\mathcal{Y})$ and a cost function $c : \mathcal{X} \times \mathcal{Y} \to \mathbb{R}^+$, Kantorovich OT seeks a joint measure $\gamma \in \mathscr{M}_+^1(\mathcal{X} \times \mathcal{Y})$ subject to $\pi_\#^{\mathcal{X}}\gamma = \mu$ and $\pi_\#^{\mathcal{Y}}\gamma = \nu$ minimizing

$$W(\mu, \nu) := \inf_\gamma \int_{\mathcal{X} \times \mathcal{Y}} c(x, y) \, d\gamma(x, y). \tag{2}$$

Note that the conditional probability distributions $\gamma_{y|x}$ specify stochastic maps from $\mathcal{X}$ to $\mathcal{Y}$ and can be considered a "one-to-many" version of the deterministic map from the Monge problem. In terms of numerical methods, the relaxed problem is a linear program that is always feasible and can be solved in $O(n^3)$ time for discrete $\mu, \nu$. Cuturi (2013) recently showed that introducing entropic regularization results in a simpler dual optimization problem that can be solved efficiently using the Sinkhorn algorithm. Based on the entropy-regularized dual problem, Genevay et al. (2016) and Seguy et al. (2017) proposed stochastic algorithms for computing transport plans that can handle continuous measures.

**Unbalanced OT.** Several formulations that extend classical OT to handle mass variation have been proposed (Chizat et al., 2015; 2018; Kondratyev et al., 2016). Existing numerical methods are based on a Kantorovich-like formulation known as *optimal-entropy transport* (Liero et al., 2018). This formulation is obtained by relaxing the marginal constraints of (2) using divergences as follows: given two positive measures $\mu \in \mathscr{M}_+(\mathcal{X})$ and $\nu \in \mathscr{M}_+(\mathcal{Y})$ and a cost function $c : \mathcal{X} \times \mathcal{Y} \to \mathbb{R}^+$, optimal entropy-transport finds a measure $\gamma \in \mathscr{M}_+(\mathcal{X} \times \mathcal{Y})$ that minimizes

$$W_{ub}(\mu, \nu) := \inf_\gamma \int_{\mathcal{X} \times \mathcal{Y}} c(x, y) \, d\gamma(x, y) + D_{\psi_1}(\pi_\#^{\mathcal{X}}\gamma | \mu) + D_{\psi_2}(\pi_\#^{\mathcal{Y}}\gamma | \nu), \tag{3}$$

where $D_{\psi_1}, D_{\psi_2}$ are $\psi$-divergences induced by $\psi_1, \psi_2$. The $\psi$-divergence between non-negative finite measures $P, Q$ over $\mathcal{T} \subseteq \mathbb{R}^d$ induced by a lower semi-continuous, convex entropy function $\psi : \mathbb{R} \to \mathbb{R} \cup \{\infty\}$ is

$$D_\psi(P|Q) := \psi'_\infty P^\perp(\mathcal{T}) + \int_{\mathcal{T}} \psi\left(\frac{dP}{dQ}\right) dQ, \tag{4}$$

where $\psi'_\infty := \lim_{s \to \infty} \frac{\psi(s)}{s}$ and $\frac{dP}{dQ}Q + P^\perp$ is the Lebesgue decomposition of $P$ with respect to $Q$. Note that mass variation is allowed since the marginals of $\gamma$ are not constrained to be $\mu$ and $\nu$. In terms of numerical

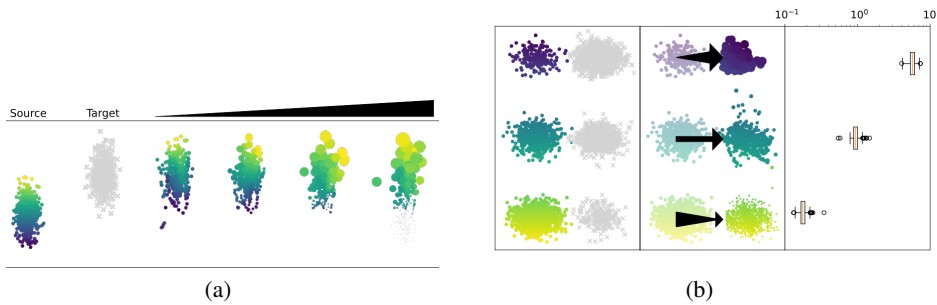

Figure 2: **Motivating examples for Unbalanced Monge OT.**

methods, the state-of-the-art in the discrete setting is a class of iterative scaling algorithms (Chizat et al., 2016) that generalize the Sinkhorn algorithm for computing regularized OT plans (Cuturi, 2013). There are no practical algorithms for unbalanced OT between continuous measures, especially in high-dimensional spaces.

## 3    SCALABLE UNBALANCED OT USING GANS

In this section, we propose the first algorithm for unbalanced OT that directly models mass variation and can be applied towards transport between high-dimensional continuous measures. The starting point of our development is the following Monge-like formulation of unbalanced OT, in which the goal is to learn a stochastic transport map and scaling factor to push a source to a target measure in a cost-optimal manner.

**Unbalanced Monge OT.** Let $c_1 : \mathcal{X} \times \mathcal{Y} \to \mathbb{R}^+$ be the cost of transport and $c_2 : \mathbb{R}^+ \to \mathbb{R}^+$ the cost of mass variation. Let the probability space $(\mathcal{Z}, \mathcal{B}(\mathcal{Z}), \lambda)$ be the source of randomness in the transport map $T$. Given two positive measures $\mu \in \mathscr{M}_+(\mathcal{X})$ and $\nu \in \mathscr{M}_+(\mathcal{Y})$, we seek a transport map $T : \mathcal{X} \times \mathcal{Z} \to \mathcal{Y}$ and a scaling factor $\xi : \mathcal{X} \to \mathbb{R}^+$ minimizing

$$L(\mu, \nu) := \inf_{T, \xi} \int_{\mathcal{X}} \int_{\mathcal{Z}} c_1(x, T(x, z)) d\lambda(z) \xi(x) d\mu(x) + \int_{\mathcal{X}} c_2(\xi(x)) d\mu(x), \tag{5}$$

subject to the constraint $T_{\#}(\xi \mu \times \lambda) = \nu$. Concretely, the first and second terms of (5) penalize the cost of mass transport and variation respectively, and the equality constraint ensures that $(T, \xi)$ pushes $\mu$ to $\nu$ exactly. A special case of (5) is the unbalanced Monge OT problem by Chizat et al. (2015), which employs a deterministic transport map (Figure 1b). We consider the more general case of stochastic (i.e. one-to-many) maps because it is a more suitable model for many practical problems. For example, in cell biology, it is natural to think of one cell in a source population as potentially giving rise to multiple cells in a target population. In practice, one can take $\mathcal{Z} = \mathbb{R}^n$ and $\lambda$ to be the standard Gaussian measure if a stochastic map is desired; otherwise $\lambda$ can be set to a deterministic distribution. The following are examples of problems that can be modeled using unbalanced Monge OT.

**Example 3.1** (Figure 2a). Suppose the objective is to model the transformation from a source measure (Column 1) to the target measure (Column 2), which represent a population of interest at two distinct time points. The transport map $T$ models the transport/movement of points from the source to the target, while the scaling factor $\xi$ models the growth (replication) or shrinkage (death) of these points. Different models of transformation are optimal depending on the relative costs of mass transport and variation (Columns 3-6).

**Example 3.2** (Figure 2b). Suppose the objective is to transport points from a source measure (1st panel, color) to a target measure (1st panel, grey) in the presence of class imbalances. A pure transport map would muddle together points from different classes, while an unbalanced transport map with a scaling factor is able to ameliorate the class imbalance (2nd panel). In this case, the scaling factor tells us explicitly how to downweigh or upweigh samples in the source distribution to balance the classes with the target distribution (3rd panel).

**Relaxation.** From an optimization standpoint, it is challenging to satisfy the constraint $T_\#(\xi\mu \times \lambda) = \nu$. We hence consider the following relaxation of (5) using a divergence penalty in place of the equality constraint:

$$L_\psi(\mu, \nu) := \inf_{T,\xi} \int_\mathcal{X} \int_\mathcal{Z} c_1(x, T(x, z)) d\lambda(z)\xi(x)d\mu(x) + \int_\mathcal{X} c_2(\xi(x))d\mu(x) + D_\psi(T_\#(\xi\mu \times \lambda)|\nu), \quad (6)$$

using an appropriate choice of $\psi$ that satisfies the requirements of Lemma C.2 in the Appendix[1].This relaxation is the Monge-like version of the optimal-entropy transport problem (3) by Liero et al. (2018). Specifically, $(T, \xi)$ specifies a joint measure $\gamma \in \mathcal{M}_+(\mathcal{X} \times \mathcal{Y})$ given by

$$\gamma(C) := \int_\mathcal{X} \int_\mathcal{Z} \mathbb{1}_C(x, T(x, z)) d\lambda(z)\xi(x)d\mu(x), \quad \forall C \in \mathcal{B}(\mathcal{X}) \times \mathcal{B}(\mathcal{Y}),$$

and by reformulating (6) in terms of $\gamma$ instead of $(T, \xi)$, one obtains the objective function for optimal-entropy transport. The main difference between the formulations is their search space, since not all joint measures $\gamma \in \mathcal{M}_+(\mathcal{X} \times \mathcal{Y})$ can be specified by some choice of $(T, \xi)$. For example, if $T$ is a deterministic transport map, then $\gamma$ is necessarily restricted to the set of deterministic couplings. Even if $T$ is sufficiently random, it is generally not possible to specify all joint measures $\gamma \in \mathcal{M}_+(\mathcal{X} \times \mathcal{Y})$: in the asymmetric Monge formulation (6), all the mass transported to $\mathcal{Y}$ must come from somewhere within the support of $\mu$, since the scaling factor $\xi$ allows mass to grow but not to materialize outside of its original support. Therefore equivalence can be established in general only when restricting the support of $\gamma$ to $supp(\mu) \times \mathcal{Y}$ as described in the following lemma, whose proof is given in the Appendix.

**Lemma 3.3.** *Let $\mathcal{G}$ be the set of joint measures supported on $supp(\mu) \times \mathcal{Y}$, and define*

$$\tilde{W}_{c,\psi_1,\psi_2}(\mu, \nu) := \inf_{\gamma \in \mathcal{G}} \int c \, d\gamma + D_{\psi_1}(\pi_\#^\mathcal{X}\gamma|\mu) + D_{\psi_2}(\pi_\#^\mathcal{Y}\gamma|\nu).$$

*If $(\mathcal{Z}, \mathcal{B}(\mathcal{Z}), \lambda)$ is atomless and $c_2$ is an entropy function, then $L_\psi(\mu, \nu) = \tilde{W}_{c_1,c_2,\psi}(\mu, \nu)$.*

Based on the relation between (3) and (6), several theoretical results for (6) follow from the analysis of optimal entropy-transport by Liero et al. (2018). Importantly, one can show the following theorem, namely that for an appropriate and sufficiently large choice of divergence penalty, solutions of the relaxed problem (6) converge to solutions of the original problem (5). The proof is given in the Appendix.

**Theorem 3.4.** *Suppose $c_1, c_2, \psi$ satisfy the existence assumptions of Proposition B.1 in the Appendix, and let $(\mathcal{Z}, \mathcal{B}(\mathcal{Z}), \lambda)$ be an atomless probability space. Furthermore, let $\psi$ be uniquely minimized at $\psi(1) = 0$. Then for a sequence $0 < \zeta^1 < \cdots < \zeta^k < \cdots$ diverging to $\infty$ indexed by $k$, $\lim_{k\to\infty} L_{\zeta^k\psi}(\mu, \nu) = L(\mu, \nu)$. Additionally, let $\gamma^k$ be the joint measure specified by a minimizer of $L_{\zeta^k\psi}(\mu, \nu)$. If $L(\mu, \nu) < \infty$, then up to extraction of a subsequence, $\gamma^k$ converges weakly to $\gamma$, the joint measure specified by a minimizer of $L(\mu, \nu)$.*

**Algorithm.** Using the relaxation of unbalanced Monge OT in (6), we now show that the transport map and scaling factor can be learned by stochastic gradient methods. While the divergence term cannot easily be minimized using the definition in (4), we can write it as a penalty witnessed by an *adversary* function $f : \mathcal{Y} \to (-\infty, \psi'_\infty]$ using the convex conjugate representation (see Lemma B.2):

$$D_\psi(T_\#(\xi\mu \times \lambda)|\nu) = \sup_f \int_\mathcal{X} \int_\mathcal{Z} f(T(x, z))d\lambda(z)\xi(x)d\mu(x) - \int_\mathcal{Y} \psi^*(f(y))d\nu(y), \quad (7)$$

where $\psi^*$ is the convex conjugate of $\psi$. The objective in (6) can now be optimized using alternating stochastic gradient updates after parameterizing $T, \xi$, and $f$ with neural networks; see Algorithm 1 [2]. The optimization

---

[1]Note that $D_\psi(T_\#(\xi\mu \times \lambda)|\nu) = 0 \nRightarrow T_\#(\xi\mu \times \lambda) = \nu$ in general since the total mass of the transported measure is not constrained; for the relaxation to be valid, $\psi(s)$ should attain a unique minimum at $s = 1$ (see Lemma C.2).

[2]We assume that one has access to samples from $\mu, \nu$, and in the setting where $\mu, \nu$ are not normalized, then samples to the normalized measures $\tilde{\mu}, \tilde{\nu}$ as well as the normalization constants $c_\mu, c_\nu$. These are reasonable assumptions for practical applications: for example, in a biological assay, one might collect $c_\mu$ cells from time point 1 and $c_\nu$ cells from time point 2. In this case, the samples are the measurements of each cell and the normalization constants are $c_\mu, c_\nu$.

procedure is similar to GAN training and can be interpreted as an adversarial game between $(T, \xi)$ and $f$:

- $T$ takes a point $x \sim \mu$ and transports it from $\mathcal{X}$ to $\mathcal{Y}$ by generating $T(x, z)$ where $z \sim \lambda$.
- $\xi$ determines the importance weight of each transported point.
- Their shared objective is to minimize the divergence between transported samples and real samples from $\nu$ that is measured by the adversary $f$.
- Additionally, cost functions $c_1$ and $c_2$ encourage $T, \xi$ to find the most cost-efficient strategy.

---
**Algorithm 1** Generative-Adversarial Framework for Unbalanced Monge OT

---
**Input:** Initial parameters $\theta, \phi, \omega$; step size $\eta$; normalized measures $\tilde{\mu}, \tilde{\nu}$, constants $c_\mu, c_\nu$.
**Output:** Updated parameters $\theta, \phi, \omega$.
**while** $(\theta, \phi, \omega)$ not converged **do**
    Sample $x_1, \cdots, x_n$ from $\tilde{\mu}$, $y_1, \cdots, y_n$ from $\tilde{\nu}$, $z_1, \cdots, z_n$ from $\lambda$;

$$\ell(\theta, \phi, \omega) := \frac{1}{n} \sum_{i=1}^{n} [c_\mu c_1(x_i, T_\theta(x_i, z_i)) \xi_\phi(x_i) + c_\mu c_2(\xi_\phi(x_i)) \qquad (8)$$

$$+ c_\mu \xi_\phi(x_i) f_\omega(T_\theta(x_i, z_i)) - c_\nu \psi^*(f_\omega(y_i)).]$$

    Update $\omega$ by gradient descent on $-\ell(\theta, \phi, \omega)$.
    Update $\theta, \phi$ by gradient descent $\ell(\theta, \phi, \omega)$.
**end while**

---

Table 1 in the Appendix provides some examples of divergences with corresponding entropy functions and convex conjugates that can be plugged into (7). Further practical considerations for implementation and training are discussed in Appendix C.

**Relation to other approaches.** The probabilistic Monge-like formulation (6) is similar to the Kantorovich-like entropy-transport problem (3) in theory, but they result in quite different numerical methods in practice. Algorithm 1 solves the non-convex formulation (6) and learns a transport map $T$ and scaling factor $\xi$ parameterized by neural networks, enabling scalable optimization using stochastic gradient descent. The networks are immediately useful for many practical applications; for instance, it only requires a single forward pass to compute the transport and scaling of a point from the source domain to the target. Furthermore, the neural architectures of $T, \xi$ imbue their function classes with a particular structure, and when chosen appropriately, enable effective learning of these functions in high-dimensional settings. Due to the non-convexity of the optimization problem, however, Algorithm 1 is not guaranteed to find the global optimum. In contrast, the scaling algorithm of Chizat et al. (2016) based on (3) solves a convex optimization problem and is proven to converge, but is currently only practical for discrete problems and has limited scalability. For completeness, in Section A of the Appendix, we propose a *new stochastic method* based on the same dual objective as Chizat et al. (2016) that can handle transport between continuous measures (Algorithm 2 in the Appendix). This method generalizes the approach of Seguy et al. (2017) for handling transport between continuous measures and overcomes the scalability limitations of Chizat et al. (2016). However, the output is in the form of the dual solution, which is less interpretable for practical applications compared to the output of Algorithm 1. In particular, while one can compute a deterministic transport map known as a barycentric projection from the dual solution, it is unclear how best to obtain a scaling factor or a stochastic transport map that can generate samples outside of the target dataset. In the numerical experiments of Section 4, we show the advantage of directly learning a transport map and scaling factor using Algorithm 1.

The problem of learning a scaling factor (or weighting factor) that "balances" measures $\mu$ and $\nu$ also arises in causal inference. Generally, $\mu$ is the distribution of covariates from a control population and $\nu$ is the distribution from a treated population. The goal is to scale the importance of different members from the

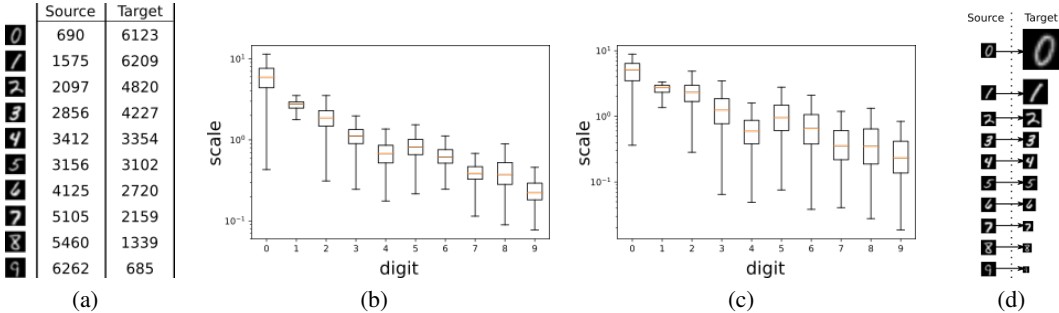

Figure 3: Learning weights on MNIST data using unbalanced OT.

control population based on how likely they are to be present in the treated population, in order to eliminate selection biases in the inference of treatment effects. Kallus (2018) proposed a generative-adversarial method for learning the scaling factor, but they do not consider transport.

## 4 NUMERICAL EXPERIMENTS

In this section, we illustrate in practice how Algorithm 1 performs unbalanced OT, with applications geared towards population modeling.

**MNIST-to-MNIST.** We first apply Algorithm 1 to perform unbalanced optimal transport between two modified MNIST datasets. The source dataset consists of regular MNIST digits with the class distribution shown in column 1 of Figure 3a. The target dataset consists of either regular (for the experiment in Figure 3b) or dimmed (for the experiment in Figure 3c) MNIST digits with the class distribution shown in column 2 of Figure 3a. The class imbalance between the source and target datasets imitates a scenerio in which certain classes (digits 0-3) become more popular and others (6-9) become less popular in the target population, while the change in brightness is meant to reflect population drift. We evaluated Algorithm 1 on the problem of transporting the source distribution to the target distribution, enforcing a high cost of transport (w.r.t. Euclidean distance). In both cases, we found that the scaling factor over each of the digit classes roughly reflects its ratio of imbalance between the source and target distributions (Figure 3b-c). These experiments validate that the scaling factor learned by Algorithm 1 reflects the class imbalances and can be used to model growth or decline of different classes in a population. Figure 3d is a schematic illustrating the reweighting that occurs during unbalanced OT.

**MNIST-to-USPS.** Next, we apply unbalanced OT from the MNIST dataset to the USPS dataset. As before, these two datasets are meant to imitate a population sampled at two different time points, this time with a large degree of evolution. We use Algorithm 1 to model the evolution of the MNIST distribution to the USPS distribution, taking as transport cost the Euclidean distance between the original and transported images. A summary of the unbalanced transport is visualized in Figure 4a. Each arrow originates from a real MNIST image and points towards the predicted appearance of this image in the USPS dataset. The size of the image reflects the scaling factor of the original MNIST image, i.e. whether it is relatively increasing or decreasing in prominence in the USPS dataset compared to the MNIST dataset according to the unbalanced OT model. Even though the Euclidean distance is not an ideal measure of correspondence between MNIST and USPS digits, many MNIST digits were able to preserve their likeness during the transport (Figure 4b). We analyzed which MNIST digits were considered as increasing or decreasing in prominence by the model. The MNIST digits with higher scaling factors were generally brighter (Figure 4c) and covered a larger area of pixels (Figure 4d) compared to the MNIST digits with lower scaling factors. These results are consistent with the observation that the target USPS digits are generally brighter and contain more pixels.

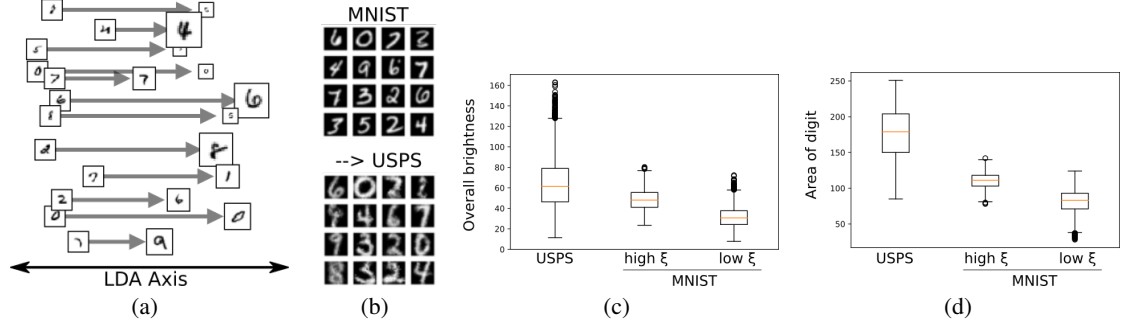

Figure 4: Unbalanced Optimal Transport from MNIST to USPS digits.

**CelebA-Young-to-CelebA-Aged.** We applied Algorithm 1 on the CelebA dataset to perform unbalanced OT from the population of young faces to the population of aged faces. This synthetic problem imitates a real application of interest, which is modeling the transformation of a population based on samples taken from two timepoints. Since the Euclidean distance between two faces is a poor measure of semantic similarity, we first train a variational autoencoder (VAE) (Kingma & Welling, 2013) on the CelebA dataset and encode all samples into the latent space. We then apply Algorithm 1 to perform unbalanced OT from the encoded young to the encoded aged faces, taking the transport cost to be the Euclidean distance in the latent space. A summary of the unbalanced transport is visualized in Figure 5a. Each arrow originates from a real face from the young population and points towards the predicted appearance of this face in the aged population. Generally, the transported faces retain the most salient features of the original faces (Figure 5b), although there are exceptions (e.g. gender swaps) which reflects that some features are not prominent components of the VAE encodings. Interestingly, the young faces with higher scaling factors were significantly enriched for males compared to young faces with lower scaling factors; 9.6% (9,913/103,287) of young female faces had a high scaling factor as compared to 18.5% (8,029/53,447) for young male faces (Figure 5c, top, $p = 0$). In other words, our model predicts growth in the prominence of male faces compared to female faces as the CelebA population evolves from young to aged. After observing this phenomenon, we confirmed based on checking the ground truth labels that there was indeed a strong gender imbalance between the young and aged populations: while the young population is predominantly female, the aged population is predominantly male (Figure 5c, bottom).

**Zebrafish embroygenesis.** A problem of great interest in biology is lineage tracing of cells between different developmental stages or during disease progression. This is a natural application of transport in which the source and target distributions are unbalanced: some cells in the earlier stage are more poised to develop into cells seen in the later stage. To showcase the relevance of learning the scaling factor, we apply Algorithm 1

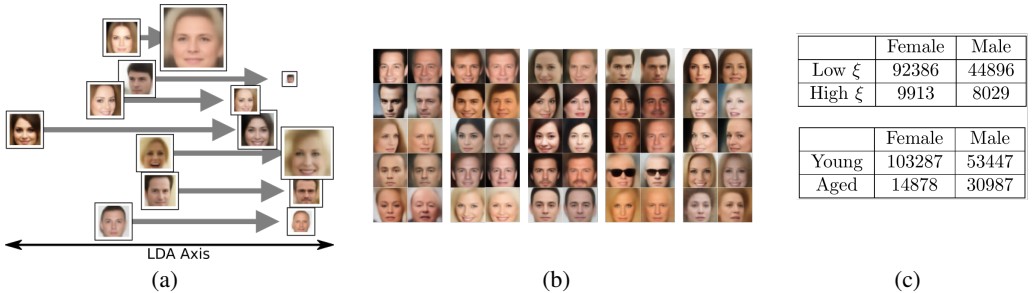

|  | Female | Male |
|---|---|---|
| Low $\xi$ | 92386 | 44896 |
| High $\xi$ | 9913 | 8029 |

|  | Female | Male |
|---|---|---|
| Young | 103287 | 53447 |
| Aged | 14878 | 30987 |

| (a) | (b) | (c) |

Figure 5: Unbalanced Optimal Transport from Young to Aged CelebA Faces.

to recent single-cell gene expression data from two stages of zebrafish embryogenesis (Farrell et al., 2018). The source population is from a late stage of blastulation and the target population from an early stage of gastrulation (Figure 6a). The results of the transport are plotted in Figure 6b-c after dimensionality reduction by PCA and T-SNE (Maaten & Hinton, 2008). To assess the scaling factor, we extracted the cells from the blastula stage with higher scaling factors (i.e. over 90th percentile) and compared them to the remainder of the cells using differential gene expression analysis, producing a ranked list of upregulated genes. Using the GOrilla tool (Eden et al., 2009), we found that the cells with higher scaling factors were significantly enriched for genes associated with differentiation and development of the mesoderm (Figure 6d). This experiment shows that analysis of the scaling factor can be applied towards interesting and meaningful biological discovery.

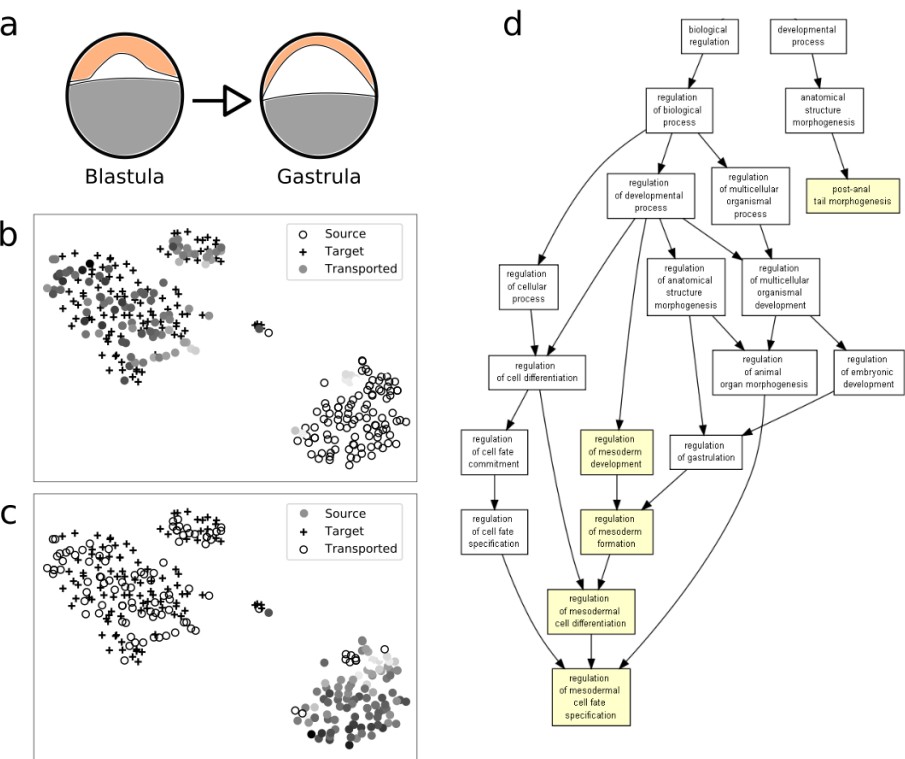

Figure 6: **Unbalanced OT on Zebrafish Single-Cell Gene Expression Data.** (a) Illustration of the blastula and gastrula stages of zebrafish embryogenesis. (b) T-SNE plot (Maaten & Hinton, 2008) of the unbalanced OT results for a subset of datapoints. The color of the transported points indicates the relative magnitude of the scaling factor (black = high, white = low). (c) Same plot as (b), where we have colored the source points instead of the transported points. (d) GOrilla output of significantly enriched processes (Eden et al., 2009) based on ranked list of enriched genes in cells with high scaling factors (black points from (c)) from a differential gene expression analysis. Processes in the graph are organized from more general (upstream) to more specific (downstream).

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

# APPENDIX

## A   DUAL STOCHASTIC METHOD

In this section, we present a stochastic method for unbalanced OT based on the regularized dual formulation of (Chizat et al., 2015), which can be considered a natural generalization of Seguy et al. (2017). The dual formulation of (3) is given by

$$\sup_{u,v} - \int \psi_1^*(-u)d\mu - \int \psi_2^*(-v)d\nu$$

subject to $u \oplus v \leq c$, where the supremum is taken over functions $u : \mathcal{X} \to [-\psi'_{1\infty}, \infty]$ and $v : \mathcal{Y} \to [-\psi'_{2\infty}, \infty]$. This is a constrained optimization problem that is challenging to solve. A standard technique for making the dual problem unconstrained is to add a strongly convex regularization term to the primal objective (Blondel et al., 2017), such as an entropic regularization term (Cuturi, 2013):

$$R_e(\gamma) = \epsilon D_{\psi_{KL}}(\gamma | \mu \otimes \nu)$$

where $\epsilon > 0$. Concretely, this term has a "smoothing" effect on the transport plan, in the sense that it encourages plans with high entropy. By the Fenchel-Rockafellar theorem, the dual of the regularized problem is given by,

$$W_{ub}^\epsilon(\mu, \nu) := \sup_{u,v} - \int \psi_1^*(-u)d\mu - \int \psi_2^*(-v)d\nu - \epsilon \int e^{(u+v-c)/\epsilon} d(\mu \otimes \nu), \qquad (9)$$

where the supremum is taken over functions $u : \mathcal{X} \to [-\psi'_{1\infty}, \infty]$ and $v : \mathcal{Y} \to [-\psi'_{2\infty}, \infty]$, and the relationship between the primal optimizer $\gamma^*$ and dual optimizer $(u^*, v^*)$ is given by

$$d\gamma^* = e^{(u^*+v^*-c)/\epsilon} d(\mu \otimes \nu). \qquad (10)$$

Next, we rewrite (9) in terms of expectations. We assume that one has access to samples from $\mu, \nu$, and in the setting where $\mu, \nu$ are not normalized, then samples to the normalized measures $\tilde{\mu}, \tilde{\nu}$ as well as the normalization constants. Based on these assumptions, we have

$$W_{ub}^\epsilon(\mu, \nu) = \sup_{u,v} -c_\mu \mathbb{E}_{x \sim \tilde{\mu}} \psi_1^*(-u(x)) - c_\nu \mathbb{E}_{y \sim \tilde{\nu}} \psi_2^*(-v(y)) - \epsilon c_\mu c_\nu \mathbb{E}_{x \sim \tilde{\mu}, y \sim \tilde{\nu}} e^{(u(x)+v(y)-c(x,y))/\epsilon}. \ (11)$$

If $\psi_1^*, \psi_2^*$ are differentiable, we can parameterize $u, v$ with neural networks $u_\theta, v_\phi$ and optimize $\theta, \phi$ using stochastic gradient descent. This is described in Algorithm 2. Note that this algorithm is a generalization of the algorithm of (Seguy et al., 2017) from classical OT to unbalanced OT. Indeed, taking $\psi_1, \psi_2$ to be equality constraints, (9) becomes

$$\sup_{u,v} \int u \, d\mu + \int v \, d\nu - \epsilon \int e^{(u+v-c)/\epsilon} d(\mu \otimes \nu),$$

which is the dual of the entropy-regularized classical OT problem.

---

**Algorithm 2** SGD for Unbalanced OT

---

**Input:** Initial parameters $\theta$, $\phi$; step size $\eta$; regularization parameter $\epsilon$; constants $c_\mu, c_\nu$ and normalized measures $\tilde{\mu}, \tilde{\nu}$
**Output:** Updated parameters $\theta$, $\phi$
**while** $(\theta, \phi)$ not converged **do**
    Sample $(x_1, y_1), \cdots, (x_n, y_n)$ from $\tilde{\mu} \otimes \tilde{\nu}$

$$\ell(\theta, \phi) := \frac{1}{n} \sum_{i=1}^{n} [c_\mu \psi_1^*(-u(x_i)) + c_\nu \psi_2^*(-v(y_i)) + \epsilon c_\mu c_\nu e^{(u(x_i)+v(y_i)-c(x_i,y_i))/\epsilon}]$$

    Update $\theta, \phi$ by gradient descent on $\ell(\theta, \phi)$
**end while**

---

The dual solution $(u^*, v^*)$ learned from Algorithm 2 can be used to reconstruct the primal solution $\gamma^*$ based on the relation in (10). Concretely, $\gamma^*$ is a transport map that indicates the amount of mass transported between every pair of points in $\mathcal{X}$ and $\mathcal{Y}$. Note that the marginals of $\gamma^*$ with respect to $\mathcal{X}$ and $\mathcal{Y}$ are not necessarily $\mu$ and $\nu$, which is where mass variation is implicitly built into the problem. Given $\gamma^*$, it is possible to also learn an "averaged" deterministic mapping from $\mathcal{X}$ to $\mathcal{Y}$. A standard approach is to take the *barycentric projection* $T : \mathcal{X} \to \mathcal{Y}$, defined as,

$$T(x) = \min_{z \in \mathcal{Y}} \mathbb{E}_{y \sim \gamma^*(\cdot|x)} d(z, y),$$

with respect to some distance $d : \mathcal{Y} \times \mathcal{Y} \to \mathbb{R}^+$. Seguy et al. (2017) proposed a stochastic algorithm for learning such a map from the dual solution, which we reproduce in Algorithm 3.

---

**Algorithm 3** Learning Barycentric Projection

---

**Input:** Learned functions $u, v$; initial $T_\theta$; distance function $d$
**Output:** Updated $T_\theta$
**while** $T_\theta$ not converged **do**
    Sample $(x_1, y_1), \cdots, (x_n, y_n)$ from $\tilde{\mu} \otimes \tilde{\nu}$

$$\ell(\theta) := \frac{1}{n} \sum_{i=1}^{n} d(x_i, T_\theta(y_i)) e^{(u(x_i)+v(y_i)-c(x_i,y_i))/\epsilon}$$

    Update $T_\theta$ by gradient descent on $\ell(\theta)$
**end while**

---

# B  SUPPLEMENT TO SECTION 3

## B.1  RELAXATION TO OPTIMAL-ENTROPY TRANSPORT

The *objectives* in (6) and (3) are equivalent if one reformulates (6) in terms of $\gamma$ instead of $(T, \xi)$, where $\gamma \in \mathcal{M}_+(\mathcal{X} \times \mathcal{Y})$ is a joint measure given by

$$\gamma(C) := \int_{\mathcal{X}} \int_{\mathcal{Z}} \mathbb{1}_C(x, T(x, z)) d\lambda(z) \xi(x) d\mu(x), \quad \forall C \in \mathcal{B}(\mathcal{X}) \times \mathcal{B}(\mathcal{Y}). \tag{12}$$

Furthermore, the formulations are equivalent if one restricts the search space of (3) to contain only those joint measures that can be specified by some $(T, \xi)$. This relation between the formulations is formalized by Lemma 3.3.

*Proof of Lemma 3.3.* First we show $L_\psi(\mu, \nu) \geq \tilde{W}_{c_1, c_2, \psi}(\mu, \nu)$. If $L_\psi(\mu, \nu) = \infty$, this is trivial, so assume $L_\psi(\mu, \nu) < \infty$. Let $(T, \xi)$ be any solution and define $\gamma$ by (12). Note by this definition that

$$\pi_\#^{\mathcal{X}} \gamma(A) = \int_{\mathcal{X}} \mathbb{1}_A(x) \xi(x) d\mu(x), \quad \forall A \in \mathcal{B}(\mathcal{X}),$$

i.e. $\xi$ is the Radon-Nikodym derivative of $\pi_\#^{\mathcal{X}} \gamma$ with respect to $\mu$. Also

$$\pi_\#^{\mathcal{Y}} \gamma(B) = \int_{\mathcal{X}} \int_{\mathcal{Z}} \mathbb{1}_B(T(x, z)) d\lambda(z) \xi(x) d\mu(x) = T_\#(\xi \mu \times \lambda)(B), \quad \forall B \in \mathcal{B}(\mathcal{Y}).$$

It follows that

$$\int_{\mathcal{X}} \left( \int_{\mathcal{Z}} c_1(x, T(x, z)) d\lambda(z) \right) \xi(x) d\mu(x) + \int_{\mathcal{X}} c_2(\xi(x)) d\mu(x) + D_\psi(T_\#(\xi \mu \times \lambda) | \nu)$$

$$= \int_{\mathcal{X} \times \mathcal{Y}} c_1(x, y) d\gamma(x, y) + \int_{\mathcal{X}} c_2(\xi(x)) d\mu(x) + D_\psi(T_\#(\xi \mu \times \lambda) | \nu)$$

(by definition of $\gamma$, linearity and monotone convergence)

$$= \int_{\mathcal{X} \times \mathcal{Y}} c_1(x, y) d\gamma(x, y) + \int_{\mathcal{X}} c_2(\frac{d\pi_\#^{\mathcal{X}} \gamma}{d\mu}) d\mu(x) + D_\psi(T_\#(\xi \mu \times \lambda) | \nu)$$

(since $\xi$ is the Radon-Nikodym derivative)

$$= \int_{\mathcal{X} \times \mathcal{Y}} c_1(x, y) d\gamma(x, y) + D_{c_2}(\pi_\#^{\mathcal{X}} \gamma | \mu) + D_\psi(T_\#(\xi \mu \times \lambda) | \nu)$$

$$= \int_{\mathcal{X} \times \mathcal{Y}} c_1(x, y) d\gamma(x, y) + D_{c_2}(\pi_\#^{\mathcal{X}} \gamma | \mu) + D_\psi(\pi_\#^{\mathcal{Y}} \gamma | \nu)$$

$$\geq \tilde{W}_{c, \psi_1, \psi_2}(\mu, \nu).$$

Since this inequality holds for any $(T, \xi)$, taking the infimum over the left-hand side yields $L_\psi(\mu, \nu) \geq \tilde{W}_{c, \psi_1, \psi_2}(\mu, \nu)$.

To show $L_\psi(\mu, \nu) \leq \tilde{W}_{c, \psi_1, \psi_2}(\mu, \nu)$, assume $\tilde{W}_{c, \psi_1, \psi_2}(\mu, \nu) < \infty$ and let $\gamma$ be any solution. By the disintegration theorem, there exists a family of probability measures $\{\gamma_{y|x}\}_{x \in \mathcal{X}}$ in $\mathcal{M}_+^1(\mathcal{Y})$ such that

$$\gamma(C) = \int_{\mathcal{X}} \int_{\mathcal{Y}} \mathbb{1}_C(x, y) d\gamma_{y|x}(y) d\pi_\#^{\mathcal{X}} \gamma, \quad \forall C \in \mathcal{B}(\mathcal{X}) \times \mathcal{B}(\mathcal{Y}).$$

Since $(\mathcal{Z}, \mathcal{B}(\mathcal{Z}), \lambda)$ is atomless, it follows from Proposition 9.1.2 and Theorem 13.1.1 in (Dudley, 2018) that there exists a family of measurable functions $\{T_x : \mathcal{Z} \to \mathcal{Y}\}_{x \in \mathcal{X}}$ such that $\gamma_{y|x}$ is the pushforward measure of $\lambda$ under $T_x$ for all $x \in \mathcal{X}$. Denoting $T(x, z) : (x, z) \mapsto T_x(z)$, then by a change of variables,

$$\gamma(C) = \int_{\mathcal{X}} \int_{\mathcal{Z}} \mathbb{1}_C(x, T(x, z)) d\lambda(z) d\pi_\#^{\mathcal{X}} \gamma, \quad \forall C \in \mathcal{B}(\mathcal{X}) \times \mathcal{B}(\mathcal{Y}).$$

By hypothesis, $\pi_\#^{\mathcal{X}} \gamma$ is restricted to the support of $\mu$, i.e. $\pi_\#^{\mathcal{X}} \gamma \lll \mu$. Let $\xi$ be the Radon-Nikodym derivative $\frac{d\pi_\#^{\mathcal{X}} \gamma}{d\mu}$. It follows from the Radon-Nikodym theorem that $(T, \xi)$ satisfy

$$\gamma(C) = \int_{\mathcal{X}} \int_{\mathcal{Z}} \mathbb{1}_C(x, T(x, z)) d\lambda(z) \xi(x) d\mu(x), \quad \forall C \in \mathcal{B}(\mathcal{X}) \times \mathcal{B}(\mathcal{Y}),$$

which is the same relation as in (12). Same as before,

$$\pi_{\#}^{\mathcal{X}}\gamma(A) = \int_{\mathcal{X}} \mathbb{1}_A(x)\xi(x)d\mu(x), \quad \forall A \in \mathcal{B}(\mathcal{X}),$$

and

$$\pi_{\#}^{\mathcal{Y}}\gamma(B) = \int_{\mathcal{X}}\int_{\mathcal{Z}} \mathbb{1}_B(T(x,z))d\lambda(z)\xi(x)d\mu(x) = T_{\#}(\xi\mu \times \lambda)(B), \quad \forall B \in \mathcal{B}(\mathcal{Y}).$$

It then follows that

$$\int_{\mathcal{X}\times\mathcal{Y}} c_1 d\gamma + D_{c_2}(\pi_{\#}^{\mathcal{X}}\gamma|\mu) + D_\psi(\pi_{\#}^{\mathcal{Y}}\gamma|\nu)$$

$$= \int_{\mathcal{X}}\left(\int_{\mathcal{Y}} c_1(x,y)d\gamma_{y|x}(y)\right)d\pi_{\#}^{\mathcal{X}}\gamma + D_{c_2}(\pi_{\#}^{\mathcal{X}}\gamma|\mu) + D_\psi(\pi_{\#}^{\mathcal{Y}}\gamma|\nu)$$

(by Fubini's Theorem for Markov kernels)

$$= \int_{\mathcal{X}}\left(\int_{\mathcal{Z}} c(x,T(x,z))d\lambda(z)\right)d\pi_{\#}^{\mathcal{X}}\gamma + D_{c_2}(\pi_{\#}^{\mathcal{X}}\gamma|\mu) + D_\psi(\pi_{\#}^{\mathcal{Y}}\gamma|\nu)$$

(by change of variables)

$$= \int_{\mathcal{X}}\left(\int_{\mathcal{Z}} c(x,T(x,z))d\lambda(z)\right)d\pi_{\#}^{\mathcal{X}}\gamma + \int c_2(\frac{d\pi_{\#}^{\mathcal{X}}\gamma}{d\mu})d\mu(x) + D_\psi(\pi_{\#}^{\mathcal{Y}}\gamma|\nu)$$

$$= \int_{\mathcal{X}}\left(\int_{\mathcal{Z}} c(x,T(x,z))d\lambda(z)\right)\xi(x)d\mu(x) + \int c_2(\xi(x))d\mu(x) + D_\psi(\pi_{\#}^{\mathcal{Y}}\gamma|\nu)$$

(by definition of $\xi$)

$$= \int_{\mathcal{X}}\left(\int_{\mathcal{Z}} c(x,T(x,z))d\lambda(z)\right)d\mu(x) + \int c_2(\xi(x))d\mu(x) + D_\psi(T_{\#}(\xi\mu \times \lambda)|\nu)$$

$$\geq L_\psi(\mu,\nu).$$

Since this inequality holds for any $\gamma$, this implies that $\tilde{W}_{c_1,c_2,\psi}(\mu,\nu) \geq L_\psi(\mu,\nu)$, which completes the proof. $\qquad\square$

Due to the near equivalence of the formulations, several theoretical results for (6) follow from the analysis of optimal entropy-transport by Liero et al. (2018), such as the following existence and uniqueness result:

**Proposition B.1.** *Suppose $L_\psi(\mu,\nu) < \infty$, $c_2$ is convex and lower semi-continuous, and $(\mathcal{Z}, \mathcal{B}(\mathcal{Z}), \lambda)$ is atomless. If (i) $c_1$ has compact sublevel sets in $\mathcal{X} \times \mathcal{Y}$ and $c'_{2\infty} + \psi'_\infty > 0$, or (ii) $c'_{2\infty} = \psi'_\infty = \infty$ then a minimizer of $L_\psi(\mu,\nu)$ exists. If $c_2, \psi$ are strictly convex, $\psi'_\infty = \infty$ and $c_1$ satisfies Corollary 3.6 (Liero et al., 2018), then the joint measure $\gamma$ specified by any minimizer of $L_\psi(\mu,\nu)$ is unique.*

*Proof of Proposition B.1.* Note that $\tilde{W}_{c_1,c_2,\psi}(\mu,\nu)$ is equivalent to $W_{c_1,c_2,\psi}(\mu,\nu)$ when $\mathcal{X}$ is restricted to the support of $\mu$. If $c_1, c_2, \psi$ satisfy (i) or (ii), they also satisfy (i) and (ii) when $\mathcal{X}$ is restricted to the support of $\mu$. By Theorem 3.3 of (Liero et al., 2018), $\tilde{W}_{c_1,c_2,\psi}(\mu,\nu)$ has a minimizer. It follows from the construction of the proof of Lemma 3.3 that a minimizer of $L_\psi(\mu,\nu)$ also exists. For uniqueness, if $\psi'_\infty = \infty$, then it follows from Lemma 3.5 of (Liero et al., 2018) and the fact that minimizers are restricted to $\mathcal{G}$ that the marginals $\pi_{\#}^{\mathcal{X}}\gamma$, $\pi_{\#}^{\mathcal{Y}}\gamma$ are uniquely determined for any solution $\gamma$ of $\tilde{W}_{c_1,c_2,\psi}(\mu,\nu)$. The uniqueness of $\gamma$ then follows from the proof of Corollary 3.6 in (Liero et al., 2018). It follows from the construction of the proof of Lemma 3.3 that the product measure generated by the minimizers of $L_\psi(\mu,\nu)$ is unique, which completes the proof. $\qquad\square$

For certain cost functions and divergences, it can be shown that $L_\psi$ defines a proper metric between positive measures $\mu$ and $\nu$, i.e. taking $c_2, \psi$ to be entropy functions corresponding to the KL-divergence and $c_1 = \log \cos_+^2(d(x,y))$, then $L_\psi(\mu, \nu)$ corresponds to the Hellinger-Kantorovich (Liero et al., 2018) or the Wasserstein-Fisher-Rao (Chizat et al., 2018) metric between positive measures $\mu$ and $\nu$.

Based on Lemma 3.3, the theoretical analysis of Liero et al. (2018), and standard results on constrained optimization, it can be shown that for an appropriate and sufficiently large choice of divergence penalty, solutions of the relaxed problem (6) converge to solutions of the original problem (5) (Theorem 3.4).

*Proof of Theorem 3.4.* Since $\zeta^k \psi(s)$ converges pointwise to the equality constraint $\iota_=(s)$, which is 0 for $s = 1$ and $\infty$ otherwise, by Lemma 3.9 in (Liero et al., 2018), we have that $\liminf_{k \to \infty} \tilde{W}_{c_1, c_2, \zeta^k \psi}(\mu, \nu) \geq \tilde{W}_{c_1, c_2, \iota_=}(\mu, \nu)$. Additionally, $\tilde{W}_{c_1, c_2, \zeta^k \psi}(\mu, \nu) \leq \tilde{W}_{c_1, c_2, \iota_=}(\mu, \nu)$ for any value of $k$ since for any minimizer $\gamma$ of $\tilde{W}_{c_1, c_2, \iota_=}(\mu, \nu)$, it holds that $\pi_\#^{\mathcal{Y}} \gamma = \nu$. Hence

$$\tilde{W}_{c_1, c_2, \iota_=}(\mu, \nu) = \int c_1 d\gamma + D_{c_2}(\pi_\#^{\mathcal{X}} \gamma | \mu) + D_{\zeta^k \psi_2}(\pi_\#^{\mathcal{Y}} \gamma | \nu) \geq \tilde{W}_{c_1, c_2, \zeta^k \psi}(\mu, \nu),$$

for all $k$. Therefore, $\lim_{k \to \infty} \tilde{W}_{c_1, c_2, \zeta^k \psi}(\mu, \nu) = \tilde{W}_{c_1, c_2, \iota_=}(\mu, \nu)$, which then by Lemma 3.3 implies the first part of the proposition.

For the second part, by the hypothesis we have that $\tilde{W}_{c_1, c_2, \iota_=}(\mu, \nu) = C < \infty$ and as a consequence $\tilde{W}_{c_1, c_2, \zeta^k \psi}(\mu, \nu) \leq C$ for all $k$. Hence, by Proposition 2.10 in (Liero et al., 2018), the sequence of minimizers $\gamma^k$ is bounded. If the assumptions of Proposition B.1 are satisfied, then the sequence $\gamma^k$ is equally tight. For assumption (ii) this follows by Proposition 2.10 in (Liero et al., 2018) and for assumption (i) this follows by the Markov inequality: for any $\lambda > 0$,

$$\gamma^k(\{(x,y) \in \mathcal{X} \times \mathcal{Y} | c_1(x,y) > \lambda\}) \leq \frac{1}{\lambda} \int c_1 d\gamma^k \leq \frac{C}{\lambda}.$$

Since $\gamma^k$ are bounded and equally tight, by an extension of Prokhorov's theorem (Theorem 2.2 of (Liero et al., 2018)), there exists a subsequence of $\gamma^k$ that is weakly convergent to some $\bar{\gamma}$. Then by lower semicontinuity, we obtain that

$$\int c_1 d\bar{\gamma} + D_{c_2}(\pi_\#^{\mathcal{X}} \bar{\gamma} | \mu) + \limsup_{k \to \infty} D_{\zeta^k \psi_2}(\pi_\#^{\mathcal{Y}} \gamma^k | \nu) \leq \tilde{W}_{c_1, c_2, \iota_=}(\mu, \nu) = C$$

Since $D_{\zeta^k \psi}(\pi_\#^{\mathcal{Y}} \gamma^k | \nu) = \zeta^k D_\psi(\pi_\#^{\mathcal{Y}} \gamma^k | \nu) \geq 0$ and $\zeta^k \to \infty$, for the left side to be finite, $D_\psi(\pi_\#^{\mathcal{Y}} \gamma^k | \nu)$ must converge to 0, so $D_\psi(\pi_\#^{\mathcal{Y}} \bar{\gamma} | \nu) = 0$ by lower semicontinuity. Therefore, $\bar{\gamma}$ is a minimizer of $\tilde{W}_{c_1, c_2, \iota_=}(\mu, \nu)$. By construction of the proof of Lemma 3.3, $\gamma^k$ is equivalent to the product measure induced by minimizers of $L_{\zeta^k \psi}(\mu, \nu)$, which implies the second part of the proposition. $\square$

## B.2 CONVEX CONJUGATE FORM OF DIVERGENCES

In this section, we present the convex conjugate form of $\psi$-divergence used to rewrite the main objective as a min-max problem.

**Lemma B.2.** *For non-negative finite measures $P, Q$ over $\mathcal{T} \subset \mathbb{R}^d$, it holds that*

$$D_\psi(P|Q) \geq \sup_{f \in \mathcal{F}} \int f dP - \psi^*(f) dQ \tag{13}$$

*where $\mathcal{F}$ is a subset of measurable functions $\{f : \mathcal{T} \to (-\infty, \psi'_\infty]\}$. Equality holds if and only if $\exists f \in \mathcal{F}$ such that (i) the restriction of $f$ to the support of $Q$ belongs to the subdifferential of $\psi(\frac{dP}{dQ})$, i.e. the Radon-Nikodym derivative of $P$ with respect to $Q$ and (ii) $f = \psi'_\infty$ over the support of $P^\perp$.*

We provide a simple proof of this result. A similar result under stronger assumptions was shown in Nguyen et al. (2008) and used by Nowozin et al. (2016) for generative modeling. A rigorous proof can be found in Liero et al. (2018).

*Proof of Lemma B.2.* Note that

$$D_\psi(P|Q) = \psi'_\infty P_\perp(\mathcal{T}) + \int_\mathcal{T} \psi\left(\frac{dP}{dQ}\right) dQ$$

$$= \psi'_\infty P_\perp(\mathcal{T}) + \int_\mathcal{T} \sup_{\xi\in\mathbb{R}} \{\xi\frac{dP}{dQ} - \psi^*(\xi)\} dQ$$

(by defintion of convex conjugate)

$$= \psi'_\infty P_\perp(\mathcal{T}) + \int_\mathcal{T} \sup_{\xi\in(-\infty,\psi'_\infty]} \{\xi\frac{dP}{dQ} - \psi^*(\xi)\} dQ$$

(by Lemma B.3) below

$$= \int_\mathcal{T} \sup_{\xi\in(-\infty,\psi'_\infty]} \{\xi dP_\perp + \xi\frac{dP}{dQ} dQ - \psi^*(\xi)dQ\}$$

$$= \sup_{f\in\mathcal{F}} \int_\mathcal{T} f dP - \psi^*(f) dQ.$$

By first-order optimality conditions, the optimal $f$ over the support of $Q$ is obtained when $\frac{dP}{dQ}$ belongs to the subdifferential of $\psi^*(f)$, or equivalently when $f$ belongs to the subdifferential of $\psi(\frac{dP}{dQ})$. It is straightforward to see that the optimal $f$ over the support of $P_\perp$ is equal to $\psi'_\infty$, which completes the proof. $\square$

**Lemma B.3.** *If $\xi > \psi'_\infty := \lim_{s\to\infty} \frac{\psi(s)}{s}$, then $\psi^*(\xi) = \infty$.*

*Proof.*

$$\psi^*(\xi) = \sup_{s\in\mathbb{R}} s\xi - \psi(s)$$

$$\geq \lim_{s\to\infty} s(\xi - \frac{\psi(s)}{s})$$

$$= \infty \text{ if } \xi > \psi'_\infty$$

$\square$

## C  PRACTICAL CONSIDERATIONS FOR NUMERICAL EXPERIMENTS

**Choice of cost functions.** Proposition B.1 gives sufficient conditions on $c_1, c_2$ for the problem to be well-posed. In practice, it is often convenient the cost of transport, $c_1$, to be some measurement of correspondence between $\mathcal{X}$ and $\mathcal{Y}$. For example, we can take $c_1(x, y)$ to be the Euclidean distance between $x$ and $y$ after mapping them to some common feature space. For the cost of mass adjustment, $c_2$, it is generally sensible to choose some convex function that vanishes at 1 (i.e. no mass adjustment) and such that $\lim_{x\to 0} c_2(x) = \lim_{x\to\infty} c_2(x) = \infty$ to prevent $\xi$ from becoming too small or too large. Any of the entropy functions shown in Table 1 are reasonable choices.

| Name | $\psi(s)$ | $D_\psi(P\|Q)$ | $\psi^*(s)$ | $\psi'_\infty$ | Activation Layer |
|---|---|---|---|---|---|
| Kullback-Leibler (KL) | $s\log s - s + 1$ | $\int \log \frac{dP}{dQ} dP - \int dP + \int dQ$ | $e^s$ | $\infty$ | N/A |
| Pearson $\chi^2$ | $(s-1)^2$ | $\int (\frac{dP}{dQ} - 1)^2 dQ$ | $\frac{s^2}{4} + s$ | $\infty$ | N/A |
| Hellinger | $(\sqrt{s} - 1)^2$ | $\int (\sqrt{\frac{dP}{dQ}} - 1)^2 dQ$ | $\frac{s}{1-s}$ | $1$ | $1 - e^s$ |
| Jensen-Shannon | $s\log s - (s+1)\log\frac{s+1}{2}$ | $\frac{1}{2}D_{KL}(P\|\frac{P+Q}{2}) + \frac{1}{2}D_{KL}(Q\|\frac{P+Q}{2})$ | $-\log(2 - e^s)$ | $\log 2$ | $\log(2) - \log(1 + e^{-s}))$ |

Table 1: Table of some common $\psi$-divergences, associated entropy functions $\psi$, and convex conjugates $\psi^*$ for Algorithm 1, partly adapted from (Nowozin et al., 2016).

**Choice of $\psi$.** In Nowozin et al. (2016), it was shown that any $\psi$-divergence could be used to train generative models, i.e. to match a generated distribution $P$ to a true data distribution $Q$. This is due to Jensen's inequality: for any convex lower semi-continous entropy function $\psi$, $D_\psi(P|Q)$ is uniquely minimized when $P = Q$, where $P, Q$ are probability measures. However, this does not generally hold when $P, Q$ are not probability measures, as illustrated by the following example.

**Example C.1.** In the original GAN paper, the discrminative objective,

$$\sup_f \int \log f(x) dP(x) - \int \log(1 - f(x)) dQ(x),$$

corresponds to $D_\psi(P|Q)$ with $\psi(s) = s\log s - (s+1)\log(s+1)$ (Nowozin et al., 2016). If $P, Q$ are probability measures, this divergence is equivalent to the Jensen-Shannon divergence and is minimized when $P = Q$. If $P, Q$ are non-negative measures with unconstrained total mass, the divergence is minimized when $P = \infty$ and $Q = 0$.

When $P, Q$ are not probability measures, we require an additional constraint on $\psi$ to ensure that divergence minimization matchces $P$ to $Q$:

**Lemma C.2.** *Suppose $P, Q$ are non-negative finite measures over $T \subseteq \mathbb{R}^n$. If $\psi(s)$ attains a unique minimum at $s = 1$ with $\psi(1) = 0$ and $\psi'_\infty > 0$, then $D_\psi(P|Q) = 0 \Rightarrow P = Q$. Otherwise, then $P \neq Q$ in general when $D_\psi(P|Q)$ is minimized.*

*Proof.* Suppose $\psi(s)$ attains a unique minimum at $s = 1$ with $\psi(1) = 0$, $\psi'_\infty > 0$, and $P \neq Q$ over a region with positive measure. It is straightforward to see by the definition in (4) that $D_\psi(P|Q) > 0$, since at least one of the two terms will be strictly positive. Therefore, the first statement holds. For the second statement, suppose either $\psi(s)$ does not attain a unique minimum at $s = 1$ or $\psi'_\infty \leq 0$. If $\psi(s)$ attains a minimum at some $s' \neq 1$, then taking $P = s'Q$ results in a divergence that is equal to or less than $P = Q$. If $\psi'_\infty \leq 0$, then letting $P = Q + P_\perp$ where $P_\perp$ is a positive measure orthogonal to $Q$ results in a divergence that is equal to or less than $P = Q$. $\square$

Table 1 provides some examples of $\psi$ corresponding to common divergences that can be used for unbalanced OT.

**Choice of $f$.** According to Lemma B.2, $f$ should belong to a class of functions that maps from $\mathcal{Y}$ to $(-\infty, \psi'_\infty]$. In practice, this can be enforced by parameterizing $f$ using a neural network with a final layer that maps to the correct range, also known as an output activation layer (Nowozin et al., 2016). Table 1 provides some examples of activation layers that can be used.

**Choice of neural architectures.** For our experiments in Section 4, we used fully-connected feedforward networks with 3 hidden layers and ReLU activations. For $T$, the output activation layer was a sigmoid function to map the final pixel brightness to the range $(0, 1)$. For $\xi$, the output activation layer was a softplus function to map the scaling factor weight to the range $(0, \infty)$.

**Gradient penalties.** The training of GANs using alternating stochastic gradient descent is not guaranteed to converge locally (Mescheder et al., 2018). Stability can be improved using regularization in the form of added instance noise (Arjovsky & Bottou, 2017) or gradient penalties (Gulrajani et al., 2017). We observed that enforcing gradient penalties on $\xi$ and $f$, while not necessary, improved the stability of Algorithm 1. A gradient penalty on $\xi$ effectively restricts the search to sufficiently smooth $\xi$, which is reasonable in practice considering that similar regions of the source measure often have similar rates of growth or contraction. A gradient penalty on $f$ changes the nature of the relaxation from (5) to (6): the right-hand side of (7) is no longer equivalent to the $\psi$-divergence, but is rather a lower-bound with a relation to bounded Lipschitz metrics (Gulrajani et al., 2017). In this case, while the problem formulation is not equivalent to optimal entropy-transport, it is still a valid relaxation of unbalanced Monge OT in (5).

**Choice of $\lambda$.** One can take $\lambda$ to be the standard Gaussian measure if a stochastic mapping is desired, similar to Almahairi et al. (2018). If a deterministic mapping is desired, then $\lambda$ is set to a deterministic distribution.

**Improved training dynamics.** Recall that the objective function for our alternating gradient updates is

$$\ell(\theta, \phi, \omega) := \frac{1}{n} \sum_{i=1}^{n} [c_\mu c_1(x_i, T_\theta(x_i, z_i)) \xi_\phi(x_i) + c_\mu c_2(\xi_\phi(x_i))$$
$$+ c_\mu \xi_\phi(x_i) f_\omega(T_\theta(x_i, z_i)) - c_\nu \psi^*(f_\omega(y_i))].$$

Early in training, $\xi_\phi$ can become very small for some $x_i$ as none of the transported samples resemble samples from the target distribution. As a result, $T_\theta$ may improve very slowly for some inputs $x_i$. One way to address this issue without changing the fixed point is to update $\xi_\phi, f_\omega$ using the above objective and update $T_\theta$ using the following objective.

$$\ell(\theta, \phi, \omega) := \frac{1}{n} \sum_{i=1}^{n} c_\mu [c_1(x_i, T_\theta(x_i, z_i)) + f_\omega(T_\theta(x_i, z_i))]$$

Note that we have omitted terms in the original objective that do not include $T_\theta$, and which therefore do not affect the gradient update. For the terms that remain, the difference is that we are rescaling the contribution of each sample $x_i$ by $1/\xi(x_i)$. As long as $\xi(x_i) > 0$, this has the effect of rescaling the gradient update of the loss function with respect to each $T_\theta(x_i, z_i)$ *without* changing the direction of the update.

