# OpenReview forum: "Scalable Unbalanced Optimal Transport using Generative Adversarial Networks"
_ICLR.cc/2019/Conference_

### Official Review · AnonReviewer2 · 2018-10-30
**good effort for scalable unbalanced OT, theoretical aspect might be problematic**

**Rating:** 6
**Confidence:** 4

**Review:**

### post rebuttal### authors addressed most of my concerns and greatly improved the manuscript and hence I am increasing my score.

Summary:

The paper introduces a static formulation for unbalanced optimal transport by learning simultaneously a transport map T and scaling factor xi .

Some theory is given to relate this formulation to unbalanced transport metrics such as Wasserstein Fisher Rao metrics  for e.g. Chizat et al 2018.

The paper proposes to  relax the constraint in the proposed static formulation using a divergence.  furthermore using a bound on the divergence , the final discrepancy proposed  is written as a min max problem between the witness function f of the divergence and the transport map T , and scaling factor xi.

An algorithm is given to find the optimal map T as a generator in GAN and to learn the scaling factor  and the witness function of the divergence with a neural network paramterization , the whole optimized with stochastic gradient.

Small experimentation on image to image transportation with unbalance in the classes is given and show how the scaling factor behaves wrt to this kind of unbalance.


Novelty and  Originality:

The paper claims that there are no known static formulations known with a scaling factor and a transport map learned simultaneously. We refer the authors to Unbalanced optimal Transport: Geometry and Kantrovich Formulation Chizat et al 2015. In page 19 in this paper Equation 2.33 a similar formulation to Equation 4 in this paper is given. (Note that phi corresponds to T and lambda to xi). This is known as the monge formulation of unbalanced optimal transport. The main difference is that the authors here introduce a stochastic map T and an additional probabilty space Z. Assuming that the mapping is deterministic those two formulations are equivalent.

Correctness:

The metric defined in this paper can be written as follow and corresponds to a generalization of the monge formulation in chizat 2015 :
L(mu,nu)= inf_{T, xi}  int   c_1(x,T_x(z) ) xi(x) lambda(z) dmu(x)  + int c_2(x_i(x)) dmu(x)
                        		 s.t T_# (xi mu)=nu
In order to get a kantorovich formulation out of this chizat et al 2015 defines semi couplings and the formulation is given in Equations 3.1 page 20.

This paper proposes to relax  T_# (xi mu)=nu with D_psi (xi \mu, \nu) and hence proposes to use:

L(mu,nu)= inf_{T, xi} int   c_1(x,T_x(z) ) xi(x) lambda(z) dmu(x)  + int c_2(x_i(x)) dmu(x)+  D_psi (xi \mu, \nu)

Lemma 3.2 of the paper claims that the formulation above corresponds to the Kantrovich formulation of unbalanced transport. I doubt the correctness of this:

Inspecting the proof of Lemma 3.2 L \geq W seems correct to me, but it is unclear what is going on in the proof of the other direction? The existence of T_x is not well supported by rigorous proof or citation? Where does xi come from in the third line of the equalities in the end of page 14? I don’t follow the equalities written at the end of page 14.

Another concern is the space Z, how does the metric depend on this space? should there be an inf on all Z?

Other comments:

- Appendix A is good wish you baselined your experiments with those algorithms.

- The experiments don’t show any benefit for learning the scaling factor, are there any applications in biology that would make a better case for this method?

- What was the architecture used to model T, xi, and f?

- Improved training dynamics in the appendix, it seems you are ignoring the weighting while optimizing on theta? than how would the weighing be beneficial ?

---

> ### Author Response · Authors · 2018-11-24
> **theoretical parts has been revised extensively and a new experiment has been added in response to feedback**
>
> Thank you for the very helpful feedback. We have heavily revised the theoretical parts for clarity and added an experiment to showcase the usefulness of the scaling factor. We believe the problematic aspects have been corrected by revising the proof for clarity and adding a citation for the step that was found questionable.
>
> - Originality
>
> Thanks for pointing out the Monge formulation by Chizat et al. (2015). We have revised Section 3 accordingly and now start by pointing out this relation at the beginning of the section.
>
> - Correctness
>
> We have rewritten the proof to improve clarity. A source of confusion may have been that we were not sufficiently clear about our choice of \Z and \lambda: in particular, the lemma holds when \lambda is an atomless measure on \Z. In this case it follows from standard results (now cited from Dudley's Real Analysis and Probability) that there exists a measurable function T_x from \Z to \Y such that \gamma_{y|x} is the pushforward of \lambda under T_x. The choice of \Z, \lambda has been clarified in the revised version of the main text, and the steps of the proof in the appendix have been rewritten.
>
> - The experiments don't show any benefit for learning the scaling factor, are there any applications in biology that would make a better case for this method?
>
> An important problem in biology is lineage tracing of cells between different stages (e.g. of development or disease progression). In these applications it is important to account for the scaling factor since the transport is not balanced; particular cells in the earlier stage are poised to develop into cells seen in the later stage, and those cells should have higher scaling factors. To showcase the relevance of learning the scaling factor for determining these poised cells, we have added an application to single-cell gene expression data taken during zebrafish embryogenesis (see the end of the paper and Appendix D). Namely, we found that the cells in the source population with higher scaling factors were significantly enriched for genes associated with differentiation and development of the mesoderm. This experiment shows that analysis of the scaling factor can be applied towards interesting and meaningful biological discovery.
>
> - What was the architecture used to model T, xi, and f?
>
> Thanks for pointing out the missing information. For our experiments, we used fully-connected feedforward networks with ReLU activations. The network for \xi has a softplus activation layer at the end to enforce non-negative values. We now describe this in Appendix C.
>
> - Improved training dynamics in the appendix, it seems you are ignoring the weighting while optimizing on theta? than how would the weighing be beneficial ?
>
> For training f and \xi, the weights are directly used. For training T, while the weights are not directly used, they are still indirectly beneficial to T (theta) because they directly affect the training of f which in turn directly affects the training of T.
>
> Thanks again for helping us improve our paper with your insightful comments.

---

> > ### Comment · AnonReviewer2 · 2018-12-03
> > **Thanks for the updates and the revision**
> >
> > I have read the rebuttal and the revision of the authors. I thank authors for updating their manuscript that improved a lot. The proof in the appendix is now much easier to follow thanks for improving it. Authors answered most of my concerns.
> >
> >  I think the new experiment Zebrafish embrogenesis is interesting and  deserves to be in the main paper you can move it to the main paper (you are allowed up to 10 pages in the main). it would be great to explore more interesting applications like this one.
> >
> > In conclusion the paper is in much better shape for publication  and hence I am increasing my score to 6.

---

> > > ### Author Response · Authors · 2018-12-03
> > > **Will move zebrafish results to main paper**
> > >
> > > We are happy to hear that you appreciated the revisions and will be happy to include the zebrafish experiment in the main paper for the camera ready. Thanks again for all of your helpful feedback.

---

### Official Review · AnonReviewer3 · 2018-11-04
**an (alternative) stochastic min-max algorithm to compute unbalanced optimal transport, using local scaling (dilatation of mass)**

**Rating:** 7
**Confidence:** 4

**Review:**

In this paper the authors consider the unbalanced optimal transport problem between two measures with different total mass. The authors introduce first the now standard Kantorovich-like formulation, which considers a coupling whose marginals are penalized to look like the two target measures. The authors introduce a second formulation in (2), somewhat a Kantorovich/Monge hybrid that involves a "random" Monge map where the target point T(x) of a point x now depends also on an additional random variable z, to desribe T(x,z). The authors also consider a local mass creation term (\xi) to weight the initial measure \mu.

The authors emphasize the interest of the 2nd formulation, which, much like the original Monge problem, has an intractable push-forward constraint. This formulation is similar to recent work on Wasserstein Autoencoders (to which is added the scaling parameter). As with WAE, this constraint is relaxed to penalize the deviation between the "random" push-forward and the desired marginal.

The authors show then that the resulting problem, which involves a transportation cost integrated both on the random variable z and on the input domain x, weighted by xi + a simple penalization for xi + a divergence penalizing the deviation between push-forward and desired marginal, can be optimized altogether by using three NN: 1 for the parameterization of T, 1 for the parameterization of \xi, and one to optimize using a function f a variational bound on the divergence. 2 gradient descents (T,\xi), 1 gradient ascent (f, variational bound).

The authors then make a link between that penalize formulation and something that resembles unbalanced transport (I say resembles because there is some assymetry, and that the type of couplings is restricted). Finally the authors show that by letting increase the penalty in front of the divergence in (6) they recover something that looks like the solution of (2).

For the sake of completeness, the authors provide in the appendix an implementation of a simple dual ascent scheme to approximate unbalanced OT inspired from previous work by Seguy'17, and show that, unlike that work, their implicit parameterization of the scaling factor \xi can help, and illustrate this numerically.

I give credit to the authors for addressing a new problem and providing an algorithmic formulation to do so. That algorithm is itself recovered from an alternative formulation of unbalanced OT, and is therefore interesting in its own right. Unfortunately, I have found the presentation rushed. I really believe the paper would deserve an extensive re-write. Everything is fairly clear until Section 3. Then, the authors introduce their main contribution.  Basically the section tries to prove two things at the same time, without really completing its job. One is to prove that "dualizing" the scaling+ random push-forward equality constraint is ok if one uses big enough regularizers (intuitive), the other that this scaled + random push-forward formulation is closely related to W_{ub}. This is less clear to me (see below).

The experiments are underwhelming. For faces they happen in latent spaces, and therefore one recovers transport between latent spaces later re-visualized through a decoder. For digits, all is fairly simple. They do not clearly mention whether this alternative UOT approach approximates UOT at all. Despite the title, there's no generation. Therefore my grade is really split between a 5 and a 6.

minor comments and questions:

- Is the reference to a local scaling (\xi) for unbalanced transport entirely new? your paper is not clear on that, and it seems to me this idea already appears in the OT literature.

- I do not understand the connexion you make with GANs. In what sense can you interpret any of your networks as generators? To me it just feels like a simultaneous optimization of various networks, yet without a clear generative purpose. Technically there may be several similarities (as we optimize on networks), but I am not sure this justifies referencing GANs in the title. Additionally, and almost mechanically, putting GAN in your paper, the reader will expect some generation results..

- Numerical benchmarks: Is the technique you propose supposed to approximate the optimal value of Unbalanced OT at all? If yes, is there a way you could compare yourselves with Chizat's approach?

- Somewhere in Lemma 3.2 the fact that you had to use an alternative definition \tilde{W} (by restricting the class of couplings) is not really clarified to the reader. Qualitatively, what does it mean that you restrict the class of couplings to have the same support as \mu? In which situations would \tilde{W} be very different from W_{ub} ? (which, if I understand correctly, only appears in (2) but not elsewhere in the paper?)

- I think it would help for the simple sake of readability to add integration domains under your \int symbols.

- T is used as a subset in Lemma 3.1, while it is used after and before as a map of (x,z)

- T(x,z) looks intuitively like a noisy encoder as in Wasserstein AEs (with, of course, the addition of your term \xi). Could you elaborate?

- I have scanned the paper but did not see how you set lambda.

---

> ### Author Response · Authors · 2018-11-24
> **Section 3 has been revised extensively and a new experiment has been added to Section 4 in response to feedback (1/2)**
>
> Thanks for your helpful comments. We have heavily revised section 3 to clarify our contributions and the relation to previous literature, taking into account the comments of all reviewers.
>
> - The experiments are underwhelming. For faces they happen in latent spaces, and therefore one recovers transport between latent spaces later re-visualized through a decoder. For digits, all is fairly simple.
>
> We agree that learning transport maps between these domains is nothing new. Rather, the main innovation in our numerical experiments is the simultaneous learning of the scaling factor that adjusts mass and accounts for class imbalances between the distributions. For example, in the MNIST experiment, the scaling factor reflects the digit imbalances between the datasets; and in the CelebA faces experiment, the scaling factor reflects the gender imbalance (i.e. predominance of males) in the aged group.
>
> To further showcase the usefulness of learning the scaling factor, we have added an application to genomics, namely based on single-cell gene expression data taken during zebrafish embryogenesis (see the end of the paper and Appendix D). When modeling transport between populations of cells from different stages of development, one needs to account for the scaling factor since the transport is not balanced: particular cells in the earlier stage are poised to develop into cells seen in the later stage and are thus overrepresented in the later stage. The new experiment shows that the scaling factor can discover these poised cells. Namely, we found that the cells in the source population with higher scaling factors were significantly enriched for genes associated with differentiation and development of the mesoderm. This experiment shows that analysis of the scaling factor can be applied towards interesting and meaningful biological discovery.
>
> -They do not clearly mention whether this alternative UOT approach approximates UOT at all.
>
> Our algorithm solves the formulation of unbalanced OT in (6). The relation to optimal entropy-transport is now clarified in Section 3; namely, the formulations are equivalent when the support of \gamma for optimal-entropy transport is subject to a support restriction. Therefore our approach does approximate unbalanced OT. Thanks for pointing out the lack of clarity.
>
> - Is the reference to a local scaling (\xi) for unbalanced transport entirely new? your paper is not clear on that, and it seems to me this idea already appears in the OT literature.
>
> Reviewer 2 provided a reference to an existing formulation that uses \xi. The relation to our work is now made clear in the revised version at the beginning of Section 3.
>
> - I do not understand the connexion you make with GANs. In what sense can you interpret any of your networks as generators?...
>
> In the revised version, the connection with GANs is clarified. We discuss how one can interpret T as a generator and Algorithm 1 as a generative-adversarial game between (T, \xi) and f, similar to a GAN. In particular,
>
> - T takes a point x ~ \lambda and transports it from X to Y by generating T(x, z) where z ~ \lambda.
> - \xi determines the importance weight of each transported point
> - their shared objective is to minimize the divergence between transported samples and real samples from \nu that is measured by the adversary f
> - cost functions c_1 and c_2 encourage T, \xi to find the most cost-efficient strategy
>
> To clarify, our paper does not contain results where images are generated from random noise; the generator in our framework is the transport map that takes a random sample from the source distribution and generates a sample in the target distribution. This is in line with previous works (e.g. unpaired image translation, CycleGAN by Zhu et al, https://arxiv.org/abs/1703.10593) where the generator in the GAN transports samples between domains rather than generating samples from random noise.
>
> - Numerical benchmarks... Is there a way you could compare yourselves with Chizat's approach?
>
> A numerical comparison of the methods would not really be meaningful. For discretized problems, we would expect the Chizat et al. method to outperform our method, since it was designed particularly for the discrete setting and solves a convex optimization problem with convergence guarantees. However, for high-dimensional/continuous problems, Chizat et al. cannot be used. Hence the methods should be considered complementary, each with its own application domains.

---

> ### Author Response · Authors · 2018-11-24
> **Section 3 has been revised extensively and a new experiment has been added to Section 4 in response to feedback (2/2)**
>
> (continued)
>
> - Somewhere in Lemma 3.2 the fact that you had to use an alternative definition \tilde{W} (by restricting the class of couplings) is not really clarified to the reader. Qualitatively, what does it mean that you restrict the class of couplings to have the same support as \mu? In which situations would \tilde{W} be very different from W_{ub} ?
>
> In the optimal entropy-transport problem (3), the objective contains a \psi-divergence that penalizes the difference between \mu and \gamma_X (the marginal of \gamma with respect to \X). Depending on which \psi-divergence is chosen, it is possible that \gamma_X has non-zero measure outside of the support of \mu. Intuitively, this means that the optimal transport scheme adds some mass to \X where there was previously no mass (since it is outside of the support of \mu) and then transports this mass to \Y. But in the asymmetric Monge formulation of (6), all the mass transported to \Y must come from somewhere within the support of \mu, since the scaling factor \xi allows mass to grow but not to materialize outside of its original support. Qualitatively, this is the effect of the support restriction. Thanks for pointing out the lack of clarity; we revised the text accordingly to make this clear to the readers.
>
> - I think it would help for the simple sake of readability to add integration domains under your \int symbols.
>
> Done; thanks for pointing this out.
>
> - T is used as a subset in Lemma 3.1, while it is used after and before as a map of (x,z)
>
> We agree that this was confusing and we adjusted the notation accordingly. Thanks for pointing this out.
>
> - T(x,z) looks intuitively like a noisy encoder as in Wasserstein AEs (with, of course, the addition of your term \xi). Could you elaborate?
>
> If one disregards the scaling factor \xi and the unbalanced aspect of our problem, both the WAE paper and our work present Monge-like formulations of the OT problem, where the objective is to learn a stochastic transport map to push one distribution to the other. In our paper, the stochastic transport map is T(x,z). In their paper, since there is a latent space, the stochastic map is the composition of the noisy encoder with the decoder map G. The notation of z is unrelated, however -- we use z as a random variable that introduces randomness into the map T, while in their work it denotes the variable in the latent space.
>
> - I have scanned the paper but did not see how you set lambda.
>
> Thanks for pointing this out. We added it to the paper in Appendix C, namely: "One can take \lambda to be the standard Gaussian measure if a stochastic mapping is desired ... if a deterministic mapping is desired, then \lambda can be set to a deterministic distribution."
>
> Thanks again for helping us improve our paper with your insightful comments.

---

> > ### Comment · AnonReviewer3 · 2018-12-09
> > **thanks for your revision**
> >
> > i have read the revised version. i also support accept. i have revised my score upwards.

---

### Official Review · AnonReviewer4 · 2018-11-11
**An adversarial formulation for unbalanced optimal transport with promising practical results and many potential applications but the theoretical part needs improvements.**

**Rating:** 6
**Confidence:** 4

**Review:**

REVIEW

The authors propose a novel approach to estimate unbalanced optimal transport between sampled measures that scales well in the dimension and in the number of samples. This formulation is based on a formulation of the entropy-transport problems of Liero et al. where the transport map, growth maps and Lagrangian multipliers are parameterized by neural networks. The effectiveness of the approach is shown on some tasks.

This is overall an ingenious contribution that opens a venue for interesting uses of optimal transport tools in learning problems (I can think for instance of transfer learning). As such I think the idea would deserve publication. However, I have some concerns with the way the theory is presented and with the lack of discussions on the theoretical limitations. Also, the theory seems a bit disconnected from the practical set up, and this should be emphasized. These concerns are detailed below.

REMARKS ON SECTION 3

I think the theoretical part does not exhibit clearly the relationships with previous literature. The formulation proposed in the paper (6) is not new and consists in solving the optimal entropy-transport problem (2) on the set of product measures gamma that are deterministic, i.e. of the form
gamma(x,y) = (id x T)_# (xi mu) for some T:X -> Y and xi : X -> R_+ (here (id x T)(x) =(x,T(x)) )
It is classical in optimal transport to switch between convex/transport plan formulation (easier to study) to non-convex/transport map formulations (easier to interpret). (As a technical note, the support restriction in Lemma 3.2 is automatically satisfied for all feasible plans, for super-linear costs c_2=phi_1).

More precisely, since the authors introduce a reference measure lambda on a space Z (these objects are not motivated anywhere, but I guess are used to allow for multivalued transport maps?), they look for plans of the form
gamma(x,y) = (pi_x x T)_# (xi mu otime lambda) where (pi_x x T)(x,z) = (x,T(x,z) and "otime" yields product measures) (it is likely that similar connections could be made with the "static" formulations in Chizat et al.).

Introduced this way, the relationship to previous literature would have been clearer and the theoretical results are simple consequences of the results in Liero et al., who have characterized when optimal solutions of this form exist. Also this contradicts the remark that the authors make that it is better to model "directly mass variation" as their formulation is essentially equivalent.

The paragraph "Relation to Unbalanced OT" is, in my opinion, incomplete. The switch to non-convex formulation introduce many differences to convex approaches that are not mentioned: there is no guarantee that a minimizer can be found, there is a bias introduced by the architecture of the neural network, ... Actually, it is this bias that make the formulation useful in high dimension since it is know that optimal transport suffers from the curse of dimensionality (thus it would be useless to try to solve it exactly in high dimension). I suggest to improve this discussion.

OTHER REMARKS
A small remark: lemma 3.1 is the convex conjugate formula for the phi-divergence in the first argument. I suggest to call it this way to help the reader connect with concepts he or she already knows. Its rigorous proof (with measurability issues properly dealt with) can be found, for instance, in Liero et al. Theorem 2.7. It follows that the central objective (8) is a Lagrangian saddle-point formulation of the problem of Liero et al., where transport plans, scalings and Lagrange multipliers are parameterized by neural networks. I generally think it is best to make the link with previous work as simple as possible.

Also, Appendix C lacks details to understand precisely how the experiments where done. It is written :
"In practice, [the correct output range] can be enforced by parameterizing f using a neural network with a final layer that maps to the correct range. In practice, we also found that employing a Lipschitz penalty on f stabilizes training."
This triggers two remarks:
- (i) how precisely is the correct range enforced? This should be stated.
- (ii) a Lipschitz penalty on f yields a class of functions which is very unlikely to have the properties of Lemma 3.1 ; in fact, this amounts to replacing the last term in (6) by a sort of "bounded Lipschitz" distance which has very different property from a f-divergence. This makes the theory of section 3 a bit disconnected from the practice of section 4.

---

> ### Author Response · Authors · 2018-11-24
> **Theoretical parts have been revised extensively and discussion improved in response to feedback**
>
> Thanks for your kind and constructive comments.
>
> We agree that section 3 could have been written more clearly, both in terms of connecting our work to existing work and in terms of motivating the material better and making it more accessible to readers.  We heavily revised section 3 based on your feedback. In particular, we now begin the section by relating our formulation to the formulation of unbalanced Monge OT by Chizat et al. (2015) and then equate the relaxed problem with the optimal entropy-transport problem by Liero et al. (2018) as per your suggestion. The point that optimal entropy-transport is the convex/transport plan version of (6) is now conveyed more clearly. What we meant by "directly modeling mass variation" is that for applications, it is often important or more intuitive to directly learn the scaling factor that indicates how much local mass dilation/contraction there is. We did not mean to imply that optimal entropy-transport does not involve mass variation; we clarified this in our revision. Additionally, the discussion comparing our approach with the existing methods based on the convex formulation has been expanded at the end of Section 3.
>
> In general, Appendix C has been expanded with more implementation details. In response to specific comments:
>
> Appendix C:
> - (i) how precisely is the correct range enforced? This should be stated.
>
> We have added to Table 1 in the Appendix some examples of final layers that show precisely how the correct range is enforced.
>
> - (ii) a Lipschitz penalty on f yields a class of functions which is very unlikely to have the properties of Lemma 3.1 ; in fact, this amounts to replacing the last term in (6) by a sort of "bounded Lipschitz" distance which has very different property from a f-divergence. This makes the theory of section 3 a bit disconnected from the practice of section 4.
>
> It should be noted that our algorithm also works without the gradient penalty on f. We added the gradient penalty since in practice this improves the stability of the training, as has also been reported in the GAN literature.
>
> In addition, we describe what the theoretical implications are of using the gradient penalty in the Appendix as follows:
>
>  "A gradient penalty on f changes the nature of the relaxation of (5) to (6): the right-hand side of (7) [convex conjugate form of divergence] is no longer equivalent to the \psi-divergence, but is rather a lower-bound with a relation to bounded Lipschitz metrics (Gulrajani 2017). In this case, while the problem formulation is not equivalent to optimal entropy transport, it is still a valid relaxation of (5) [unbalanced Monge OT]."
>
> Thanks again for helping us improve our paper with your insightful comments.

---

> > ### Comment · AnonReviewer4 · 2018-12-08
> > **revision**
> >
> > I have read the revised manuscript. I found that the revised version is more clear, more precise and reflects better the quality of the underlying ideas. It is better at distinguishing between what was known and what is new. I have also appreciated the new numerical experiment. For these reasons and also the ones I mentioned in my previous review, I suggest acceptance and update my score to 6.

---

### Meta-Review · Area_Chair1 · 2018-12-18
**An interesting algorithm for unbalanced optimal transport**

**Confidence:** 4
**Recommendation:** Accept (Poster)

**Metareview:**

After revision, all reviewers agree that this paper makes an interesting contribution to ICLR by proposing a new methodology for unbalanced optimal transport using GANs and should be accepted.